# Sexual Dimorphisms, Anti-Hormonal Therapy and Cardiac Arrhythmias

**DOI:** 10.3390/ijms22031464

**Published:** 2021-02-02

**Authors:** Virginie Grouthier, Melissa Y. Y. Moey, Estelle Gandjbakhch, Xavier Waintraub, Christian Funck-Brentano, Anne Bachelot, Joe-Elie Salem

**Affiliations:** 1Department of Endocrinology, Diabetes and Nutrition, Centre Hospitalier Universitaire de Bordeaux, Haut Leveque Hospital, F-33000 Bordeaux, France; virginie.grouthier@chu-bordeaux.fr; 2Department of Cardiovascular Disease, Vidant Medical Center/East Carolina University, Greenville, NC 27834, USA; mel.moey@gmail.com; 3APHP, Pitié-Salpêtrière Hospital, Institute of Cardiology, Centre de Référence des Maladies Cardiaques Héréditaires, Institute of Cardiometabolism and Nutrition (ICAN), UPMC Univ Paris 06, INSERM 1166, Sorbonne Universités, F-75013 Paris, France; estelle.gandjbakhch@aphp.fr (E.G.); xavier.waintraub@aphp.fr (X.W.); 4INSERM, CIC-1901, AP-HP, Pitié-Salpêtrière Hospital, Regional Pharmacovigilance Center, UNICO-GRECO Cardio-Oncology Program, Department of Pharmacology and Clinical Investigation Center, CLIP2 Galilée, Sorbonne Université, F-75013 Paris, France; christian.funck-brentano@aphp.fr; 5AP-HP, Pitié-Salpêtrière Hospital, IE3M, and Centre de Référence des Maladies Endocriniennes Rares de la Croissance, and Centre de Référence des Pathologies Gynécologiques Rares, Department of Endocrinology and Reproductive Medicine, Sorbonne Université, F-75013 Paris, France; anne.bachelot@aphp.fr; 6Cardio-Oncology Program, Department of Medicine and Pharmacology, Vanderbilt University Medical Center, Nashville, TN 37232, USA

**Keywords:** sex, anticancer drugs, atrial fibrillation, QT, ventricular arrhythmias

## Abstract

Significant variations from the normal QT interval range of 350 to 450 milliseconds (ms) in men and 360 to 460 ms in women increase the risk for ventricular arrhythmias. This difference in the QT interval between men and women has led to the understanding of the influence of sex hormones on the role of gender-specific channelopathies and development of ventricular arrhythmias. The QT interval, which represents the duration of ventricular repolarization of the heart, can be affected by androgen levels, resulting in a sex-specific predilection for acquired and inherited channelopathies such as acquired long QT syndrome in women and Brugada syndrome and early repolarization syndrome in men. Manipulation of the homeostasis of these sex hormones as either hormonal therapy for certain cancers, recreational therapy or family planning and in transgender treatment has also been shown to affect QT interval duration and increase the risk for ventricular arrhythmias. In this review, we highlight the effects of endogenous and exogenous sex hormones in the physiological and pathological states on QTc variation and predisposition to gender-specific pro-arrhythmias.

## 1. Introduction

The QT interval of the electrocardiogram, which represents the total duration of ventricular depolarization and repolarization [1], is measured from the initiation of the QRS complex to the termination of the T wave. The QTc interval is the QT interval corrected for heart rate and normally ranges between 350 and 450 milliseconds (ms) in men and 360 to 460 ms in women [2] (Figure 1). Values below and above these limits are suggestive of underlying channelopathies [3]. There are many widely used rate correction formulas to calculate QTc, namely, Bazett’s, Fridericia’s, Framingham’s and Hodge’s, all with their own merits and disadvantages, that normalize the QT interval obtained at a given cycle length to 60 beats/min [4]. The QTc interval is a widely used marker of ventricular arrhythmia risk. Indeed, prolonged QTc is associated with a higher risk of ventricular arrhythmias, especially torsade de pointes (TdP) which can be fatal with sudden cardiac death (SCD) [5]. Compared with normal QTc values, the risk of TdP increases 2- to 3-fold if the QTc interval exceeds the 500 ms threshold and even more so if it exceeds 550 ms [6].

Ventricular repolarization, reflected by the QT interval, is a heterogenous process which can be affected by a variety factors with the main determinants including heart rate, autonomic activity, age and gender in both acquired and congenital conditions [4]. Reviewed extensively elsewhere [4], ventricular repolarization is largely modulated by potassium outward channels where perturbance from acquired conditions such as structural disorders (myocardial ischemia, bundle branch block, antiarrhythmics, electrolyte disturbances) or congenital disorders with loss- or gain-of-function mutations affecting the potassium channels can result in QT prolongation (congenital long QT syndrome (LQTS)) or shortening (short QT syndrome, J wave syndromes), respectively, and ultimately pro-arrhythmias [7].

The prevalence and incidence of cardiac pro-arrhythmic disorders are often influenced by sex due to specific hormonal effects on the QT interval with a predilection of certain channelopathies to women vs. men (Figure 1). Acquired long QT syndrome (LQTS), which is mainly drug-induced (diLQTS), is often secondary to defects in ventricular repolarizing currents (rapid delayed rectifier potassium current [I_Kr_]) associated with excessive action potential duration (APD) lengthening and is most prevalent in women [3,8]. Congenital LQTS (cLQTS) is an inherited arrhythmia syndrome due to genetic dysfunction of cardiac ion channels that alter the action potential [9]. The follow-up of a large cLQTS population of 1710 patients during a median of 7,1 years highlighted that female sex is an independent predictor of life-threatening events (HR: 1.70; 95% CI: 1.00 to 2.88; *p* = 0.048) [10]. In contrast, Brugada syndrome (BrS) and early repolarization syndrome (ERS) are more frequent in men and are due to a transmural voltage gradient created by an imbalance in the cardiac ion currents involved in phase 1 of the action potential, specifically a loss of function of the sodium (I_Na_) or calcium (I_Ca_) inward depolarization current and a gain of function of the transient outward potassium current (Ito) [11]. This leads to the presence of an action potential notch described as a “J wave” or slurring of the terminal portion of the QRS, classic to the “J wave syndromes” of BrS and ERS. Disturbances in these specific cardiac ion channels have been localized to the right ventricular outflow tract and commonly the inferior left ventricular wall in BrS and ERS, respectively [12,13]. LQTS, BrS and ERS favor different forms of malignant ventricular arrhythmias—for example, TdP for LQTS and ventricular tachycardia for BrS and ERS that may lead to ventricular fibrillation (VF) and, if sustained, SCD [3,8,14,15]. This sex-specific prevalence of LQTS, BrS and ERS is partly explained by the fact that androgens shorten APD and QT duration, while its deprivation has opposite effects [16,17].

The current literature on arrhythmogenic substrates and differences in susceptibility to certain ventricular arrhythmias between the different sexes is still lacking. For example, the mechanism behind a lower incidence of SCD in women in comparison to men despite controlling for concomitant structural heart disease (ischemia, heart failure) [18] or the sex-specific biological basis of the tendency for women to develop acquired LQTS more than men with use of the class IA and III antiarrhythmic therapies than men [19] has yet to be clearly elucidated and remains under investigation. In addition, women are often underrepresented in major randomized clinical trials such as with antiarrhythmic therapies, primary prevention of SCD or cardiac resynchronization therapies, making it challenging to generalize study findings and their clinical applications to both sexes. Now, with the increased use of exogenous hormonal therapies in certain cancers, recreational therapy, family planning and transgender treatment, understanding the role of sexual dimorphism in the risk for cardiac pro-arrhythmias is of absolute importance. In this detailed review, we highlight the importance of understanding the physiology of sexual dimorphism through a detailed discussion of the effects of sex-specific hormones in the physiological and pathological states and exogenous use on QTc variation and predisposition to pro-arrhythmias (Table 1).

## 2. Sex Differences in QT Duration and TdP Risk

Women have a higher risk of TdP which occurs twice as often than in men [20,21,22]. A longer QTc duration in healthy women compared to healthy men noted by Bazett in 1920 [23] has since been observed by numerous studies, suggesting that sex hormones play an essential role in this phenomenon [24,25,26,27,28,29,30,31,32,33,34]. In fact, there is no difference in the QTc interval between genders during childhood, but QTc shortens in men at the time of puberty. This gender difference in QTc values decreases with age due to progressive QTc lengthening in males concordant with the decrease in testosterone levels over decades until QTc reaches the level of women around the age of menopause [32,34,35]. Although variations in the QTc interval remain within normal ranges in the healthy population, these observations led to the hypothesis that cardiac repolarization is influenced by sex steroid hormones resulting in a sexual dimorphism of arrhythmias and, in particular, in the lengthening of QTc [17,35,36].

## 3. Role of Estrogen and Progesterone during the Menstrual Cycle, and Hormonal Substitution in Menopause in QTc Variation

It has been well established that estradiol and progesterone have opposite effects on ventricular repolarization with lengthening of the QTc interval by estradiol and shortening of the QTc interval by progesterone [35]. In women, estradiol increases at the beginning of the follicular phase, followed by an exponential rise with ovulation. Progesterone levels subsequently increase during the luteal phase with a subsequent rapid decrease in estradiol and progesterone during menstruation. Fluctuations in estradiol and progesterone levels during the menstrual cycle have therefore been implicated in the increased risk of ventricular arrhythmias and particularly diTdP, though this remains inconsistent in the literature [37,38,39,40].

DiLQTS occurred more frequently in women during the ovulation period on ibulitide, a known QT prolonging anti-arrhythmic drug that blocks IKr [41]. During the menstrual cycle in healthy women, the QTc interval measured in the follicular phase concomitantly with increased estradiol level was higher than the QTc interval measured in the luteal phase [35,42]. Rodriguez et al. have shown that progesterone (r = −0.40) and the progesterone-to-estradiol ratio (r = −0.41) were inversely correlated with ibutilide-induced QT prolongation [39]. Nakagawa et al. also reported that the QTc interval is approximately 10 ms shorter in the luteal phase, resulting from progesterone’s QTc shortening effect offsetting estrogen’s QTc prolongation effect [43]. Conversely, in 2016, Dogan et al. found no difference in the QTc interval between menstruation and ovulation periods [40] and Hulot et al. did not demonstrate a relationship between the level of estrogen and the QTc interval duration (*p* = 0.92) [38]. A recent study highlighted the role of some progesterone metabolites which have known inherent estrogenic activity on cardiac repolarization [44]. The 6β-hydroxyprogesterone and 16α-hydroxyprogesterone metabolites were independent predictors of QTc interval prolongation. These metabolites, although principally produced during fetal life, have been demonstrated in vitro to inhibit IKr and thus to prolong ventricular repolarization [44,45]. In contrast, progesterone activates the slow delayed rectifier potassium current (IKs) and inhibits the L-type calcium current (ICa-L), counterbalancing IKr inhibition and overcoming the QTc prolonging effects of its metabolites (Figure 2).

Menopause is defined as a permanent cessation of menstruation for at least 1 year that typically occurs at 50 years of age. Several studies that analyzed the effect of hormone replacement treatment (HRT) (estrogens + progestins or estrogens alone) on ventricular repolarization in postmenopausal women remain conflicting [46,47,48]. One would expect a prolongation of the QTc interval with estrogen treatment counterbalanced by a QT shortening when added to progestins with androgenic activity in postmenopausal women without structural heart disease [49,50].

A German study compared the impact of HRT on QTc duration among 22 postmenopausal women without any hormonal treatment (control group) vs. 16 postmenopausal women with only estrogens treatment and 22 postmenopausal women treated by a combination of estrogens and progestins (no details were provided on the progestins used). The length of QTc was significantly higher in patients receiving estrogens alone compared to the control group (423 ± 28 vs. 398 ± 31 ms, *p* < 0.05); however, there was no difference in QTc duration with the addition of progestins compared with controls [50]. A similar observation was also confirmed several years later by a US study [51], where the QTc interval increased in postmenopausal women after estrogen therapy (386 ± 11.6 ms prior to HRT and 398 ± 17 ms with HRT, *p* < 0.05) with no difference observed in patients receiving combination HRT (QTc 394 ± 10.1 vs. 388 ± 26.4 ms, NS). Data on specific doses of estrogen or combined hormone therapy were not available. In addition, Tisdale et al. have recently shown in a randomized, double-blind, placebo-controlled crossover study that oral progesterone attenuated ibutilide’s effect on QTc prolongation [52]. In castrated rabbits implanted with sustained release pellets of progesterone or placebo, a significant decrease in the incidence of dofetilide-induced TdP (27% vs. 61%, respectively, *p* = 0.049), bigeminy (36% vs. 74%, *p* = 0.03) and trigeminy (18% vs. 57%, *p* = 0.01) [53] was demonstrated in the progesterone-exposed group. Overall, estrogens therefore play a role in lengthening QTc, while progesterone appears to be protective by limiting QTc prolongation.

## 4. QTc Variation in Hypoestrogenic and Hyperandrogenic States in Women

The role of female hormonal imbalance particularly in ovarian or adrenal diseases can also impact the QTc interval and subsequent risk for ventricular arrhythmias. Selected therapies used for treatment of these pathologies may also affect the QTc interval.

Premature ovarian insufficiency (POI), which occurs in 1% of women, is a condition where the ovaries stop functioning normally before age 40 and is characterized by menstrual disturbance with elevated gonadotropins and low estradiol [54]. Treatment for POI typically involves HRT with estrogens and progestins until the normal age of menopause. In addition to the lengthening effect of estradiol and the shortening effect of progesterone on QTc duration, the follicle-stimulating hormone (FSH) also appears to be a factor that is positively correlated with the duration of ventricular repolarization [36]. The rationale for this complex hormonal system regulating QTc is supported by the fact that the RNA of gonadotropin receptors is expressed in the myocardium [55]. Knowledge of cardiac repolarization in women with POI is, however, poorly documented by a few studies on Turner syndrome.

Turner syndrome (TS) is a well-known cause of POI and is associated with a risk of heart disease [56]. TS is a classic situation with low progesterone and estradiol levels and a high FSH level, which could therefore play a role in QTc duration. Nielsen et al. illustrated the case of a 58-year-old woman with TS absent of long QT gene mutations, who developed significant QT prolongation (465 to 600 ms) 12 h after a single dose of 300 mg of intravenous amiodarone for AF [57]. Seventeen hours later, she had a cardiac arrest with VF treated by immediate cardioversion without additional amiodarone treatment with normalization of her ECG within a few days. In a recent publication, 112 genetically proven TS patients were followed for an average of 7.0 ± 5.1 years after the first ECG [58]. At least one QTc prolonging medication was prescribed in 81 (74%) patients; however, none had documented ventricular arrhythmias in comparison to 112 age-matched controls without TS. QTc was significantly shorter in matched controls using either Bazett’s or Hodge’s formula compared to TS patients (424 ± 16 vs. 448 ± 28 ms, *p* < 0.0001; and 414.8 ± 16 vs. 424.2 ± 20 ms; *p* = 0.0002, respectively). Unfortunately, no information on hormone therapy was available.

Several studies have analyzed the QTc interval in patients with TS and POI, though with many limitations [59,60]. These studies were from a small number of patients with only limited data on HRT and most without hormonal evaluation. Moreover, QTc was frequently calculated with Bazett’s formula that does not take into account the higher heart rate in TS as with the use of Hodge’s formula, which is the preferred QTc correction in these patients [56]. In a recent cross-sectional study, Atici et al. demonstrated that 38 patients with TS had significantly longer QTc than 35 controls (respectively, QTc 411.9 ± 22.7 vs. 392.1 ± 13.2 ms, *p* < 0.001), with the major limitation being the lack of data on karyotype or hormonal treatment [60]. This lengthening of the QTc interval had already been highlighted by Bondy et al. as early as 2006 [61]. In their study population, 36% of the TS patients (after puberty and discontinuation of HRT) had a QTc above the reference range in comparison to only 4% of the controls (440 ms; *p* = 0.0001).

In 2017, clinical practice guidelines for the care of women with TS suggested that a resting ECG with QTc measurement should be performed at the time of diagnosis, in order to avoid prolonging QTc drugs in patients with evidence of baseline QTc prolongation. If deemed necessary, an ECG should be performed 1–2 weeks after initiation of QT-prolonging drugs [56]. Recently this year, a larger study of 359 women found no difference in QTc duration compared to the general population. These authors described a lower prevalence of QTc prolongation in patients with TS compared to previous studies, which may be a result of a different cutoff value used for the definition of QTc prolongation (450 for girls and 460 ms for women, in comparison to a cutoff of 440 ms in other studies) [62]. Another possible explanation could be linked to the karyotypic formula of the patients (monosomia or mosaic). Although the association of karyotype with Y chromosomal material and shortening of QTc was recently demonstrated by Harrahill et al. [58], this association has been rarely studied and the outcomes remain contradicting [61,62]. Finally, a high prevalence of mutations in the major LQTS genes in women with TS and prolonged QTc have already been discussed in 2013 [63]. Indeed, 88 TS patients with a prolonged QTc > 432 ms by Bazett’s formula were found to have seven mutations in major LQTS genes (SCN5A and KCNH2) and one in a minor LQTS gene (KCNE2).

Hyperandrogenic states in women such as polycystic ovary syndrome (PCOS) or congenital adrenal hyperplasia (CAH) also highlight the important role of the hormone level balance on QTc duration. The androgen level is, in fact, another major determinant of the duration of ventricular repolarization, where a significant decrease in the testosterone level lengthens the QTc interval in men [17]. The results of studies regarding the influence of testosterone on cardiac repolarization in women, however, are still contradictory. Indeed, no correlation has been found between the testosterone level and QTc interval by Zhang et al. [64] or Nakagawa et al. [43]. However, in these studies, women did not have hyperandrogenic states and correlation between the two had been researched based on quartiles of testosterone normal ranges. It was very likely that the impact on the QTc was greater in the pathological situation of excess testosterone.

PCOS is the most common endocrinopathy affecting reproductive age women (8–13% of population) defined by the Rotterdam PCOS Diagnostic Criteria (two of oligo- or anovulation, clinical and/or biochemical hyperandrogenism or polycystic ovaries on ultrasound), after exclusion of related disorders such as CAH, Cushing disease, hyperprolactinemia and androgen-secreting tumor [65]. In women with PCOS, a negative correlation between testosterone levels and QTc has actually been highlighted in comparison to healthy women [66]. In a prospective case–control study, the QTc interval was compared in 119 PCOS women with hyperandrogenism and 64 age-matched healthy women without PCOS and without hyperandrogenism [67]. Among PCOS patients with an elevated testosterone level compared to controls (2.7 ± 2.1 vs. 1.4 ± 1.7 nmol/L, respectively *p* = 0.01), QTc duration was shorter in PCOS patients than in controls (401 ± 61 vs. 467 ± 61 ms; *p* = 0.007). This was inversely related to the plasma levels of testosterone (Spearman −0.45, *p* = 0.005). However, these results were not observed by Akdag et al. (QTc of patients with PCOS: 414 ± 33.7 vs. control group: 417 ± 32.9 ms, *p* = 0.082) [68]. Patients with PCOS had higher serum testosterone than control subjects (respectively, 95.6 ± 55.9 ng/dL vs. 49.6 ± 41.3 ng/gL, *p* < 0.001) but also a higher estradiol level, although no significant changes in ECGs were observed.

In recent years, multiple studies have focused on the chronic low-grade inflammation process in visceral adipose tissue and its significant role in the pathophysiological mechanisms of PCOS disease [69]. Inflammatory cytokines appear to have direct effects on ventricular repolarization and may play a role in lengthening of the QT duration via cytokine-mediated changes in potassium channel expression [70,71,72]. As PCOS involves hyperandrogenism and an increased inflammatory state, the analysis of QTc duration is complex; inflammatory status should therefore be considered as another concurrent variable of the phenotype of PCOS and an arrhythmic risk factor.

Treatment of PCOS may require recombinant FSH administration to improve fertility. The possible lengthening effect of FSH on QTc has already been highlighted [36] and it would be worth evaluating whether the short QTc associated with this disease would be corrected by FSH administration, especially with a significant increase in the estradiol level due to stimulation of ovulation. Although it did not reach a pathological level, in a recent study that assessed the effect of supraphysiological estradiol level changes (estradiol peak at 1656 ± 878 pg/mL) occurring during in vitro fertilization treatment on ventricular repolarization in 59 patients [73], mean QTc intervals increased significantly after ovarian hyperstimulation treatment (411.9 ± 23.7 before treatment and 420.7 ± 23.3 ms during ovarian hyperstimulation; *p* = 0.007).

Less common than PCOS, CAH is an autosomal recessive disease due to 21-hydroxylase deficiency and is characterized by cortisol deficiency associated with excess androgen [74]. Abeshira et al. studied the combined influence of several sex steroid hormones and gonadotropins on QTc interval duration in a prospective case–control study of 68 patients with CAH (44 women and 24 men) and 81 age-matched healthy controls (55 women and 26 men) [36]. 17-OH-progesterone, progesterone, the progesterone/estradiol ratio and total testosterone were higher in women with CAH than in female controls (*p* < 0.05). CAH women had shorter QTc intervals compared to female controls (404 ± 2.0 vs. 413 ± 2.1 ms; *p* = 0.001), but the QTc interval in CAH women was not statistically different to CAH men (404.7 ± 3.7 ms) or healthy men (396 ± 2.8 ms). According to univariable analysis, QTc duration was negatively correlated with progesterone (r = −0.29; *p* = 0.01), progesterone/estradiol ratio (r = −0.38; *p* = 0.001) and total testosterone (r = −0.2; *p* = 0.01) in women. All of these small variations in QTc duration are likely of minimal clinical relevance when not associated with other risk factors for TdP; however, they illustrate the important but complex role of sex hormones in ventricular repolarization.

## 5. Exogenous Hormonal Therapy and its Effect on QTc Variation

Altering sex hormone receptor pathways through endocrine therapies may modify ventricular repolarization and can be proarrhythmogenic. Hormonal therapy such as contraception or breast cancer therapy in women and exogenous testosterone or androgenic deprivation therapy (ADT) for prostate cancer in men is frequently and increasingly used, and its effects on ventricular repolarization have garnered significant interest. Indeed, male acquired hypogonadism or ADT blunts the 10–20 msec difference in QTc between men and women [8].

### 5.1. Exogenous Hormonal Therapy in Women

Various hormonal oral contraceptive pills exist such as combined estrogen–progestin or progestin-only pills. Depending on the progestin used, oral contraceptive pills have variable antiandrogenic action. Oral contraceptive generations (first, second and third; e.g., nortestosterone derivatives) are classified by progestin decreasing androgenic potency, while the fourth (or other) generations are antiandrogenic pills. For instance, levonorgestrel is a second-generation progestin with high androgenic potency, while desogestrel, norgestimate and gestodene, which are third-generation progestins, have intermediate androgenic potency, and chlormadinone acetate, cyproterone acetate, dienogest or drospirenone, which are not nortestosterone derivatives, have a predominantly antiandrogenic effect. Drospirenone also has anti-mineralocorticoid activity.

In a US retrospective study analyzing 410,782 ECGs from 34,676 women on oral contraceptive pills, QTc was found to be shorter in users of first and second generations than non-users (*p* < 0.0001) and longer in users of the fourth generation than non-users (*p* < 0.0001) [75]. In a recent cohort of 498 healthy non-menopausal women, drospirenone was associated with increased sotalol-induced QTc prolongation ( + 6–7 ms) as compared to controls or levonorgestrel [16,76]. Salem et al. also confirmed this finding using the European pharmacovigilance database with a disproportionality analysis showing higher reporting rates of suspected diLQTS, ventricular arrhythmias and cardiac arrest for drospirenone vs. levonorgestrel [76].

In cancer therapy, endocrine therapies are drugs that interfere with hormone-activated pathways to slow cancer progression, most commonly used for the treatment of breast cancer. A recent review described in detail the cardiac arrhythmia effects of hormonal cancer therapy [8]. Many such pharmaceutical agents prolong the QTc interval by interacting with cardiac ion channels, mainly blocking I_Kr_ and enhancing the late sodium current (I_Na-L_) [77]. Prolongation of ventricular repolarization with several oncologic drugs has been shown to be notably mediated by inhibition of the α-subunit of phosphoinositide 3-kinase [78,79]. Selective estrogen receptor modulators (SERMs) and aromatase inhibitors (AIs) are cornerstone endocrine drugs used for hormone-sensitive breast cancer treatment. AIs block peripheral conversion of testosterone into estradiol, leading to decreased estradiol and increased testosterone levels. Kurokawa et al. recently showed that ablation of circulating levels of estradiol in aromatase knockout mice blunted the QT prolonging effect of E-4031, an I_Kr_ blocker [80]. Moreover, it has been shown that high-dose anastrozole leads to QT shortening in dogs. Results from animal models should be extrapolated with caution in humans because of potential differences in the composition of cardiac ionic channels among both species [81]. In a recent pharmacovigilance study, SERMs were found to be associated with a higher proportion of adverse drug reaction reports related to LQT, TdP and ventricular arrhythmias compared with AIs of suspected adverse drug reaction reports [82] (Figure 2).

Although the QTc lengthening due to oral contraceptive pills or cancer therapy remain moderate and do not appear clinically significant, possible combinations with other QT-prolonging drugs should be taken into consideration, as well as the diagnosis of cLQTS. For instance, association between tamoxifen and antidepressants such as serotonin reuptake inhibitors, particularly paroxetine, frequently used in breast cancer, induces a significant increase in the QTc interval [83].

### 5.2. Exogenous Hormonal Therapy in Men

Exogenous testosterone and dihydrotestosterone (DHT) can shorten QTc duration in vivo and decrease the APD in preclinical models [8,35]. The ECG pattern of a longer repolarization duration (longer QT interval and lower T wave amplitude) in women has been well established to differ from men as a result of the androgenic effect of testosterone. In men with hypogonadism who received exogenous intramuscular injection of testosterone, there was a significant negative linear relationship observed between the QT interval and testosterone level, whereby QTc was longer (median QTc 363 ms, IQR 357 to 384 ms) during lower levels of testosterone and shorter (median QTc 352 ms, IQR 340 to 363 ms) during higher levels of testosterone [84].

In a larger study that looked at the ECG parameters of two randomized controlled clinical trials assessing testosterone replacement in opioid-induced androgen deprivation patients (Testosterone and Pain, TAP) and the effect on progression of subclinical atherosclerosis in older men (Testosterone Effects on Atherosclerosis in Aging Men, TEAAM), testosterone reduced the age-related increase in QTc duration in men [85]. Specifically, within the TAP trial [86], a change in QTc was negatively associated with a change in total testosterone (r = −0.24, *p* = 0.036) and in the TEAAM trial [87], there was a decrease in QTc by −6.3 ms (*p* < 0.001).

Tribulus terrestris belongs to the plant family of Zygophyllaceae that has been used since ancient times for a multitude of effects, particularly as an aphrodisiac, for improvement in sexual dysfunction and for its anti-inflammatory and anti-oxidant properties [88]. Several studies have demonstrated in different animal models that Tribulus terrestris increases the levels of testosterone as well as its active metabolite (DHT) and dehydroepiandrosterone sulphate [88]. Particularly among primates, there was a mean 51% and 32% increase in testosterone and DHT, respectively, with Tribulus terrestris ingestion in comparison to normal primate subjects [88]. In castrated rats, administration of Tribulus terrestris increased testosterone levels by 25%. The effect of the testosterone and DHT increase is suspected to be secondary to the predominant steroidal glycoside (saponins) furostonal of Tribulus terrestris, protodioscin, which has a role in androgen production through either direct hormonal receptor binding or alteration of the steroidal enzyme metabolism [88]. Indeed, in patients with hypogonadism, protodioscin treatment has been shown to increase levels of testosterone and luteinizing hormones.

However, the effects of exogenous chronic anabolic–androgenic steroid (AAS) use, such as oxymetholone, stanozolol and nandrolone, are more complex as they can have variable effects on the ECG pattern due to their different androgenic and anabolic properties (anabolic-to-androgen ratio) and interactions with gonadotropins, which are dependent on the chemical structure of the compound studied [35]. AASs are testosterone derivatives with the primary function to have a higher anabolic-to-androgen ratio; for example, oxymetholone has an anabolic-to-androgen ratio of 9, while stanozolol has an anabolic-to-androgen ratio of 30, in comparison to that of testosterone of 1 [89]. Thus, chronic AAS use can cause a significant increase in non-testosterone androgen plasma levels which results in a negative feedback mechanism on endogenous testosterone secretion by the testes due to suppression of the hypothalamic–pituitary–gonadal axis. QTc was even found to be prolonged in some studies of chronic AAS use in male bodybuilders [35].

Acquired male hypogonadism in patients receiving ADT for prostate cancer therapy has been reported as a risk factor for acquired LQTS and TdP [17,85,90,91,92] and normalization of the testosterone level by HRT attenuates the modification of QTc duration in hypogonadal males [85,93,94]. In a case series of seven consecutive men prospectively admitted for TdP, we showed that all had acquired hypogonadism and that normalization of testosterone levels shortened QTc and prevented recurrence of TdP [17]. The effects of testosterone on the QT interval were also demonstrated in a small randomized controlled study of men where ibutilide-induced QTcF prolongation was reversed with subdermal testosterone in comparison to placebo [95]. Similarly, in extreme cases with severely decreased total, free and bioavailable testosterone levels which were demonstrated in a retrospective analysis from a TdP registry, patients receiving ADT, particularly leuprolide and bicalutamide for prostate cancer, had prolonged QTc (QTc 530–620 ms) and suffered TdP [91].

The data on the mechanism of the effect of hormonal changes specific to gender on the ventricular repolarization and APD have not entirely been elucidated; however, they have been summarized in recent reviews [8,35] (Figure 2). Specifically, the change in morphology and shortening of the ventricular APD was shown to be related to a higher I_To_ (transient outward potassium current) density in male ventricular epicardial cells than in female cardiomyocytes in dogs [96]. In addition, in rodent-based preclinical studies, testosterone has been suggested to shorten the APD by enhancement of the repolarizing I_Kr_ and I_Ks_ currents and by decreasing the depolarizing L-type calcium current (I_Ca-L_) [96]. In a translational study that identified an association of enzalutamide (a strong androgen receptor antagonist) with diLQTS and high fatality rates, enzalutamide was found to significantly prolong the APD in cardiomyocytes from induced pluripotent stem cells from men [90]. With exposure to both acute (15 min) and chronic (5 h) enzalutamide, the APD was significantly prolonged and corresponded to an inhibition of I_Kr_ and an enhancement of I_Na-L_, which was abrogated by DHT administration [90].

## 6. Role of Sex and Inherited Channelopathies

### 6.1. Inherited Channelopathies in Women

In addition to acquired LQTS which is often drug-induced [21,97], cLQTS is another cause of palpitations, syncope, seizures, cardiac arrest and SCD. cLQTS are inherited primary arrhythmia syndromes due to dysfunction of cardiac ion channels that alter the action potential [9]. The prevalence of cLQTS is evaluated in approximately 1 out of every 2000 healthy live births. Although there has not been a direct association of all gene mutations with cLQTS, several mutations linked to the disorder have been identified, the most common of which has been found in the potassium channel KCNQ1 (cLQTS1) and hERG (human Ether-à-go-go Related Gene, cLQTS2) genes. cLQTS1 affects 40–55% of cLQTS individuals. The KCNQ1 gene, expressed in the cell membrane of cardiomyocytes, encodes the α-subunit of a voltage-gated potassium channel which contributes to generate the I_Ks_ current. cLQTS2 represents 30–45% of the cLQTS population. hERG (KCNH2) codes for a voltage-gated pore forming the α-subunit of the inwardly rectifying potassium channel which contributes to generate the I_Kr_ current. The incidence of life-threatening events is lowest for cLQTS1 compared to cLQTS2. In comparison, cLQTS3 affects approximately 5–10% of the population and is associated with more than 300 gain-of-function mutations of the SCN5A gene. It is considered the most lethal among the cLQTS phenotypes as malignant VAs can occur at rest and during sleep.

Sex hormones have varying effects on the calcium (I_Ca,L_) and potassium (I_Kr_, I_Ks_ and I_K1_ channel) currents (Figure 2) in both genomic- and non-genomic-regulated pathways. In the conventional genomic pathway, sex hormones bind to transmembrane sex hormone receptors that translocate into the nucleus and regulate expression of target cardiac ion channel genes. For example, pre-treatment with testosterone in vitro has been shown to increase the expression of Ica-L, resulting in a shorter QTc interval [98], while estrogen decreases the expression of Iks and Ikr, resulting in lengthening of the QT interval [99]. Sex hormones have also been demonstrated to exert effects on ventricular repolarization through non-genomic regulation, particularly activation of the endothelial nitric oxide (eNOS) and mitogen activating protein kinase (MAPK)-associated intracellular pathways [100]. The non-transcriptional alterations of specific cardiac ion channels by sex hormones in the acute setting with preclinical models have further provided evidence of their effects on regulation of the QT interval and predisposition to arrhythmias. In an animal model of guinea pig ventricular cardiomyocytes designed to represent fluctuations in the hormonal balance during the menstrual cycle in women, acute exposure to progesterone resulted in APD shortening due to phosphoinositide 3-kinase (PI3K)/Akt-dependent eNOS pathway enhancement of IKs and inhibition of ICa-L currents [101]. The same group has also previously shown a similar non-genomic effect of testosterone via the PI3K/Akt-dependent eNOS pathway [102].

Female sex is an independent risk factor for syncope and SCD in patients with cLQTS during adulthood and cLQTS adult women have a lengthened QTc interval vs. men [6,103,104]. Moreover, female sex has been recently determined to be an independent predictor of life-threatening events for cLQTS patients [10]. Anneken et al. reported that QTc was shortened during pregnancy in patients with cLQTS2 as compared to the post-partum period [105]. In cLQTS2 women, the risk for cardiac events (syncope, TdP and SCD) is more than 4-fold increased during the postpartum (due to the rapid decrease in progesterone) vs. pre-pregnancy period, while the arrhythmogenic risk during pregnancy is reduced by 50%. Recently, Odening et al. reported a well-documented case of a 37-year-old female suspected cLQTS2 patient who had normalization of her QTc interval during pregnancy and the postpartum period when she breastfed or when a progesterone-releasing intrauterine device was used [106]. This same group had concordantly shown that progesterone could successfully prevent induction of ventricular arrhythmia in a rabbit model of LQTS2 [107]. Although exogenous use of hormonal therapies such as medroxyprogesterone acetate (MPA) for contraception may exert similar antiproliferative effects to progesterone on the endometrium, it does not appear to affect the non-genomic eNOS pathway of cardiac ion current regulation and in patients with cLQTs, and use of MPA may increase the risk for TdP and SCD [108].

### 6.2. Inherited Channelopathies in Men

In comparison, male sex, higher testosterone levels and use of androgen agonists are risk factors for ERS and BrS (Figure 1) favoring ventricular tachycardia and VF [109]. Indeed, in comparison to LQTS which has a strong female prevalence, ERS and BrS are both predominantly seen in men (60–90%), particularly of younger age, which has been demonstrated to be related specifically to higher testosterone levels [110]. ERS and BrS, referred to collectively by some as “J wave syndromes” [11], although each with independently unique ECG characteristics, are typically manifested as specific J wave patterns associated with malignant ventricular arrhythmias. The J or Osborn wave first described in 1953 is the positive deflection from the isoelectric line observed during hypothermia [111] but is commonly seen in the young male population.

The classic ECG pattern of type 1 BrS shows a “coved” ST segment—J-point elevation with concave ST segment elevation merging into a negative or isoelectric symmetric T wave. This pattern is typically observed in leads V1 and V2 placed in the third or second intercostal space. The literature describing the QTc interval in BrS is limited to the first original description of the ECG and clinical phenotypes by Brugada and Brugada. The QTc in BrS is generally perceived to be within the normal gender-specific range in patients with the SCN5A mutation, the gene that encodes the α-subunit of the cardiac sodium channel leading to the I_Na_ current [15]. A shorter QTc, however, has been observed in BrS with loss-of-function mutations in the genes encoding I_Ca-L_ [112]. The direct effect of testosterone on BrS has also been shown by surgical castration and ADT [113] in patients with prostate cancer, which resulted in the disappearance of the classical BrS ECG phenotype, suggesting that testosterone may modulate the early phase of ventricular repolarization.

In ERS, which is highly prevalent in the male population (90%), the QTc is significantly shorter in males than in females and is suspected to be due to the androgenic effects of testosterone on the repolarization duration [14]. An inferolateral early repolarization pattern (J-point elevation of ≥0.1 mV in ≥2 leads in inferior [II, III, aVF] and/or lateral [I, aVL, V4-V6] leads) in ERS is of particular clinical significance due to its proclivity for SCD in young males. This channelopathy has been associated with heritable mutations linked to the KCNJ8, CACNA1C, CACNB2b and CACNA2D1 subunit genes of the I_Ca-L_ channel and the ATP-sensitive potassium channel subunit gene (ABCC9). Among 2755 male patients from the Health 2000 study [110], subjects who exhibited an inferolateral early repolarization pattern were found to have a higher total testosterone level than those with normal ECG patterns (17.81 ± 6.24 vs. 15.73 ± 6.10 nmol/l, *p* < 0.0001). This was also observed in a tertile-level-dependent manner, whereby the ERS ECG pattern increased 2-fold between each successive tertile, as well as in a recent Turkish study of 179 healthy male patients [114]. These results further corroborated earlier findings which observed a decrease in prevalence in the ERS ECG pattern in middle-aged and older men vs. younger men [115].

### 6.3. Atrial Fibrillation and Sex Hormones

AF is the most common cardiac dysrhythmia with a high prevalence in men and with increasing age which has been proposed to be linked with testosterone deficiency as the incidence of hypogonadism is present in approximately 38% of men older than 45 years of age [116]. Among 275 of 1251 men from the Framingham Study who developed incidental AF, a significant interaction between the levels of testosterone and AF risk was observed. Particularly among older men between the ages of 55 and 69 years old and in octogenarians and above, there was a respective 1.3-fold and 3.5-fold increase in AF risk with each 1 standard deviation decrease in the testosterone level [116]. In a large national cohort study from 76,639 patients with low testosterone identified from the Veterans Administrations Corporate Data Warehouse, patients who were administered testosterone replacement therapy (TRT) and had normalized testosterone levels had a significantly lower risk for AF in comparison to those who received TRT with persistently low levels of testosterone (HR 0.90, 95% CI 0.81–0.99, *p* = 0.0255) and patients who were not treated with TRT (HR 0.79, 95% CI 0.70–0.89, *p* = 0.0001) [117]. The association between a decrease in testosterone levels and risk for AF development has been suspected due to altered atrial electrical monitoring from increased inflammation and metabolic disorders such as obesity, hyperlipidemia and hypertension [116,118].

Testosterone deficiency and AF have also been demonstrated in a rat preclinical model [119]. In orchiectomized male Sprague-Dawley rats, electrically stimulated repetitive atrial responses were resolved with the administration of testosterone suggested to be related to decreased calcium leakage from the sarcoendoplasmic reticulum as a result of normalization of binding of the FK506-binding protein (FKBP12.6) to the ryanodine receptor type 2 (RyR2) [119]. On the contrary, however, in aged rabbits receiving TRT, there was an increased incidence of early afterdepolarizations and arrhythmogenesis in the pulmonary vein and left atrium via enhanced adrenergic activity [120].

## 7. QTc Variation in Transgender Individuals

An interesting research study model to assess the effects of sexual dimorphism on cardiac repolarization is the transgender person. Indeed, sex hormone levels of the desired gender in a transgender person will reach the same level of those of a cisgender person, i.e., a transwoman (birth-assigned male, i.e., gender that is assumed on the basis of the phenotype/physical sex characteristics present at birth whose gender identity is female) will have very low testosterone and high estradiol levels, and a transman (birth-assigned female whose gender identity is male) will have increased testosterone levels and lowered estradiol and progesterone levels [121].

Wamboldt et al. reported ECG analyses of two cases of transwomen who used estrogen therapy associated with an antiandrogen drug [122]. In the first case series, a healthy 26-year-old transwoman without heart disease was treated with oral estradiol (4 mg), cyproterone acetate (25 mg) and finasteride (1 mg). Her ECG showed a sinus rhythm with frequent premature ventricular contractions. In the second case, a 52-year-old transwoman was treated by estradiol hemihydrate (2 mg), progesterone (100 mg) and buserelin acetate (1 mg/mL 3 sprays per day). She had hypertension, type 1 diabetes mellitus and smoking history but no heart disease. Hormonal evaluation was unavailable. Manifestation of AF was diagnosed but occurred during hypoglycemia. Her ECG showed a sinus rhythm with premature atrial complexes. Unfortunately, QTc measurements were not specified. Recently, the case of a 63-year-old transman with cardiac arrest was also reported [123]. This patient was treated by testosterone for about 20 years and developed the Brugada pattern.

To date, very few data are available on cardiac repolarization in this population. Antwi-Amoabeng et al. reported a 3.68% incidence of cardiac arrhythmias in a retrospective cohort of 16,555 transgender adults hospitalized for gender-affirming surgeries. Among this cohort, arrhythmia occurred significantly more in transmen than in transwomen (*p* = 0.0029) and those who were significantly older [124]. The most common arrythmia was AF (2.87%, lower than the reported 4% incidence in the general surgery population) followed by atrial flutter and ventricular tachycardia. VF was an infrequent arrythmia (<0.09%). To our knowledge, there are no data available on QTc modifications in this population to date, but ECG monitoring before and during hormonal treatment appears appropriate.

## 8. Conclusions

Large studies have shown that diLQTS and TdP are more prevalent in women than in men, contrasting with ERS and BrS which are more frequent in men. This can be, in part, explained by circulating testosterone levels since male hypogonadism or ADT blunts the ERS and Brugada ECG features and the 10–20 msec difference in QTc between men and women. Susceptibility to diLQTS is variable in women depending on the menstrual phase, with higher risk during the follicular phase when the estradiol/progesterone ratio is high vs. the luteal phase characterized by high progesterone levels. Exogenous hormonal therapies can alter the clinical penetrance of these cardiac pro-arrhythmia phenotypes, particularly in congenitally predisposed patients, which emphasizes the need for careful monitoring and risk assessment in these patients.

## Figures and Tables

**Figure 1 ijms-22-01464-f001:**
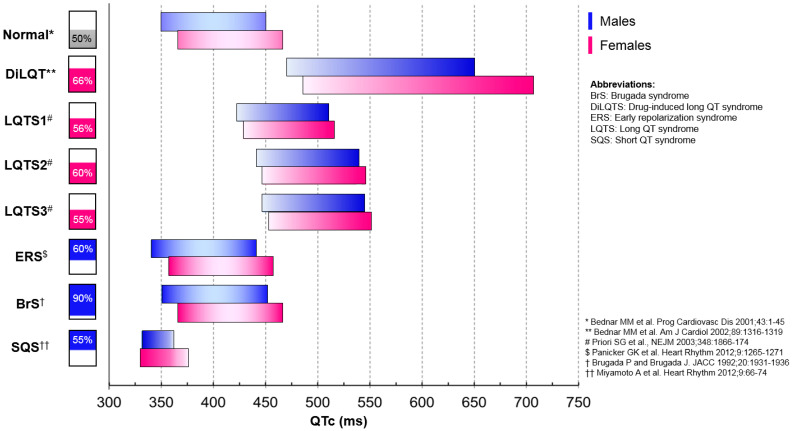
QTc duration as a function of gender (blue for males and pink for females) and main cardiac proarrhythmogenic channelopathies. Horizontal rectangular bars represent the range of QTc for normal and proarrhythmogenic channelopathies. Darker gradient color within the horizontal rectangular bars represents an increased risk for arrhythmia. Vertical rectangular bars represent the prevalence for these conditions as a function of gender. DiLQTS: drug-induced long QT syndrome, LQTS(X): congenital long QT syndrome type (X), ERS: early repolarization syndrome, BrS: Brugada syndrome, SQS: short QT syndrome.

**Figure 2 ijms-22-01464-f002:**
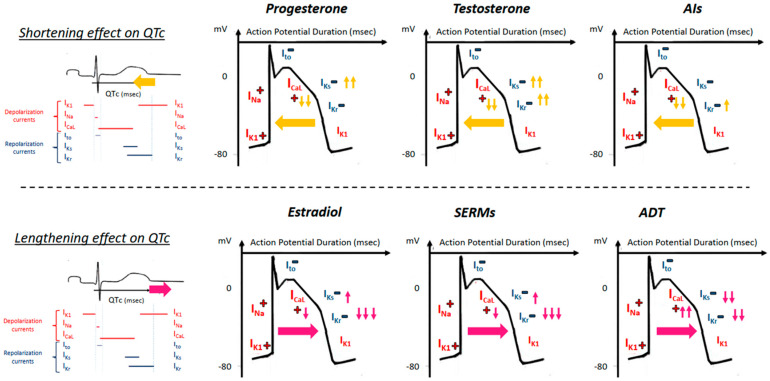
Representation of the influence of different endogenous sex hormones and hormonal therapy on ventricular repolarization. Abbreviations: ADT—androgen deprivation therapy, AIs—aromatase inhibitors, SERMs—selective estrogen receptor modulators.

**Table 1 ijms-22-01464-t001:** Effects of endogenous sex hormones and hormonal therapies on ventricular repolarization based on gender.

**Men**	
**Testosterone**	QTc shortening in normal men and in hypogonadal men receiving testosterone QTc lengthening in hypogonadal men
**Estradiol**	No influence within physiological ranges Possible QTc lengthening in transwomen
**Progesterone**	No influence within physiological ranges
**Gonadotropins**	QTc lengthening
**Androgen deprivation therapy**	QTc lengthening
**Anabolic steroid**	QTc shortening for exogenous testosterone, tribulus terrestris Variable effects on QTc for other exogenous anabolic-androgenic steroids
**Women**	
**Progesterone**	QTc shortening
**Estradiol**	QTc lengthening particularly when progesterone levels are low
**Testosterone**	No influence within physiologic ranges QTc shortening within supraphysiological testosterone levels
**Gonadotropins**	QTc lengthening
**Contraceptive pills**	QTc shortening with first and second contraceptive generation QTc lengthening with anti-androgenic contraceptive pills
**Hormone replacement therapy**	QTc lengthening with HRT with estrogens alone No influence with HRT combining estrogens and progestins
**Selective estrogen receptor modulators**	QTc lengthening
**Aromatase inhibitors**	QTc shortening

## Data Availability

The data presented in this study are available on request from the corresponding author.

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
