# Peer review of "Sexual Dimorphisms, Anti-Hormonal Therapy and Cardiac Arrhythmias"

_ijms, 2021, doi:10.3390/ijms22031464_

Round 1
Reviewer 1 Report
Comments on: Sexual dimorphisms, anti-hormonal therapy and cardiac arrhythmias:
Authors: Virginie Grouthier, Melissa Y.Y. Moey, Estelle Gandjbakhch, Xavier Waintraub, Christian Funck-Brentano, Anne Bachelot, Joe-Elie Salem*
This is a review article that describes an important disease mechanism of QT interval prolongation and an important role for sexual dimorphism in its pathogenesis and/or maintained effects. This is timely and therefore reinforces the lack of progress in the field of cardiac arrhythmias and underlying molecular mechanisms. Beyond the merits and significance of the dialogue, the article has some limitations that if addressed will significantly improve the quality of theme.
My suggestions-comments are as follows:
- Abbreviations have to be better utilized. For example, ventricular arrhythmias is abbreviated? Excessive abbreviations make it somewhat difficult to maintain the flow of the theme.
- The introduction could be better developed by initially introducing the differences between inherited (mutations in ion channels), and/or acquired (i.e., drug induced or acquired in disease states).
- Despite the significance of the case studies, it alters the flow of the manuscript. Is this a case study or a review article? These should be removed or integrated better as it adds nothing significant to the context of the manuscript as it stands now.
- It would be important to the theme of the article to emphasize why the field will benefit from studying the unappreciated role of sexual dimorphisms and how future studies could be improved and inform rational therapeutic interventions with significant implications in patients.
- The manuscript will benefit from tables that illustrates the multiple changes in ion channel mechanisms and sex related signaling pathways. Additionally, cartoon representations should be seriously considered as these are likely to better illustrate the authors’ reasoning and ideas.
- The authors need to avoid reiterating well-established concepts. Sexual dimorphisms and arrhythmias have been widely reported and comprehensively studied. Several sections of the manuscript are excessively long and should be abridged to focus on the narrative the authors are trying to share with the scientific community. The article is densely written but with limited interpretation of concepts. Therefore, the article needs to be structured better.
Reviewer 2 Report
The topic is of great interest and the review is preatty comprehensive. Yet, major revisions are needed both from a conceptual and a stylistical point of view.
General comments:
Introduction:
- it seems appropriate to also refer to the recent paper by Mazzanti et al (Mazzanti A, et al. Interplay Between Genetic Substrate, QTc Duration, and Arrhythmia Risk in Patients With Long QT Syndrome. J Am Coll Cardiol. 2018; 71(15):1663-1671) where the group demonstrated in the largest LQTS population described so far that female sex is an independent predictor of life-threatening events during follow up (HR: 1.70; 95% CI: 1.00 to 2.88; p=0.048).
- The pathophysiology of the J wave syndromes must be revised. Notably, the paramount feature in these disorders is not action potential shortening, rather an imbalance in ionic currents involved in the phase 1 of AP leading to a notch in the shape of the epicardial AP and consequently to a transmural voltage gradient, that is typically appreciable at the surface ECG where Ito current is more prominent, namely RVOT in BrS and the inferior wall (mostly) in ERS. Indeed, preclinical studies showed that testosterone and dihydrotestosterone levels effect the Na, Ca and K (especially Ito) channels forming the action potential dome. In addition to modulate ionic current involved in phase 1 of the cardiac AP, dihydrotestosterone also modulates AP duration affecting both AP50 and AP95 and leading to AP shortening.
Case 1: some important clarifications are needed.
- As already reported in figure 2 label, it should be specified in the text as well that the normal QTc referred to dates to 1995 (as compared to AVB and TdP occurring on 2019, based on the data reported on the ECG traces)
- The ECG trace showing complete AV block associated with severe QTc prolongation is dated April 20th, 2019, while the ECG trace showing TdP is dated April 22nd, 2019. Therefore, the AVB as cause of TdP cannot be defined as acute. What happened in the intercurrent timeframe between the two ECG?
- Despite the suboptimal resolution of the trace, it seems like the QTc in figure 2 B is a little bit overestimated: the reviewer measures a QTc (Fredericia) of respectively 574 msec in DII and 569 msec in V5, and a QTc (Bazett) of respectively 535 msec in DII and 528 msec in V5.
- The patient was also suffering It is now well established that the systemic inflammatory status (Lazzerini PE et al, Systemic inflammation as a novel QT-prolonging risk factor in patients with torsades de pointes. Heart. 2017;103(22):1821-1829) may be an important cofactor in reducing the repolarization reserve, therefore more details about the history of the disease and the systemic inflammatory markers at the time of the event (at least C reactive protein levels) should be provided.
- The exact values of plasmatic potassium, calcium and magnesium should be provided.
- An ECG trace after pacemaker implantation and hormonal supplementation therapy should be provided. The author disagrees with the sentence stating that paced rhythm completely precludes the evaluation of QTc. They can either provide the QTc and the QRS duration value, or the QTc, the QRS duration value and the JTc value.
Case 2: some important clarifications are needed.
- in the opinion of the reviewer, it should be clarified that the ECG in figure 3 B cannot be considered typical for BrS, since a diffuse intraventricular conduction delay is present as well as diffuse abnormalities in ventricular repolarization suggesting RV as well as LV acute overload (LVEF reported to be 30%). Also, the ECG was obtained 3 hours after several DC shock and during hypothermia (please specify the temperature). Although the authors stated that the ECG was “compatible” not typical, still the abovementioned clarification are needed. Notably, none of the ECG reported in figure 3 includes the ECG filters (both high pass and low pass filters should be reported), that should instead be included to assure a proper signal filtering.
The presentation about PCOS should be ampliated considering that hyperandrogenism is also typically associated with an increased systemic inflammatory status. In view of the recent evidence about the role of systemic inflammation as a novel QT-prolonging risk factor (Lazzerini PE et al. Cardiac Arrest Risk During Acute Infections: Systemic Inflammation Directly Prolongs QTc Interval via Cytokine-Mediated Effects on Potassium Channel Expression. Circ Arrhythm Electrophysiol. 2020;13(8): e008627), further substantiated during the SarSCov2 pandemic (Lazzerini PE, Boutjdir M, Capecchi PL. COVID-19, Arrhythmic Risk, and Inflammation: Mind the Gap! Circulation. 2020;142(1):7-9), this may contribute to explain why the relationship between testosterone levels, QTc and arrhythmic risk in PCOS in complex (inflammatory status should be considered as another variable concurring to the phenotype and the arrhythmic risk).
Specific comments:
Page 1, line 47: The meaning of QTc interval should be clearly stated as well as a brief mention of the possibility of using different correction formulae.
Page 3: the authors should justify the usage of Fridericia’s correction formula as opposed to the most largely used in the clinical setting, namely the Bazett’s formula. It would be rather preferable to use both in order to have a clinical referral value as well (most clinical studies assessing the risk of TdP use the Bazett’s formula, although the Fridericia’s one is the most used to assess the impact on drug on the QTc at a pre-marketing phase). Notably, most, if not all the paramount paper assessing the relationship between QTc length and ventricular arrhythmias, have used the Bazett’s correction formula, despite its very well-known limitations.
Page 5, line 159: please rephrase “cardiac conduction system” is not appropriate
Page 6, line 148: please reformulate the sentence, with QTc instead of “It”
Page 11, line 451: the sentence starting with “Each” needs to be rephrased.
Page 11, line 464: when describing the effects of sex hormones on cardiac ionic channels, a brief mechanicistic overview and distinction between genomic and non-genomic effects would be appropriate. These additional insights may also provide the reader a better comprehension and prospective on the expected timeline of sex hormone induced effects on cardiac AP and on the arrhythmic risk. For instance, the non-genomic pathway mediated by eNOS is thought to provide an important contribution to progesterone’s mediated effects on AP (Nakamura H, Kurokawa J, Bai CX, Asada K, Xu J, Oren RV, Zhu ZI, Clancy CE, Isobe M, Furukawa T. Progesterone regulates cardiac repolarization through a nongenomic pathway: an in vitro patch-clamp and computational modeling study. Circulation. 2007 Dec 18;116(25):2913-22). It may also concur to explain why some exogenous progestinic such as medroxyprogesterone acetate, lacking direct androgenic activity (albeit potentially able to interfere with androgenic receptor) as well as the capability to activate cardiac eNOS, have been reported to be proarrhythmic in the setting of LQTS (Giudicessi JR, Brost BC, Traynor KD, Ackerman MJ. Potential depot medroxyprogesterone acetate-triggered torsades de pointes in a case of congenital type 2 long QT syndrome. Heart Rhythm. 2012 Jul;9(7):1143-7.).
Page 11, line 470: As already mentioned, it seems appropriate to also refer to the recent paper by Mazzanti et al (Mazzanti A, et al. Interplay Between Genetic Substrate, QTc Duration, and Arrhythmia Risk in Patients With Long QT Syndrome. J Am Coll Cardiol. 2018; 71(15):1663-1671) where the group demonstrated in the largest LQTS population described so far that female sex is an independent predictor of life-threatening events during follow up (HR: 1.70; 95% CI: 1.00 to 2.88; p=0.048).
Page 11, line 491: please rephrase the sentence, it is too long and contains too many repetitions. Also, please be more accurate concerning the location of the electrocardiographic findings (leads and intercostal spaces).
Page 11, line 493: please rephrase the sentence, it is too long
Page 12, line 522: additional data from the Framingham study could be reported to clarify the findings. Indeed, the Framingham study showed an association between AF incidence and reduced total testosterone levels in men aged 55 years and above, with the strongest association seen in men ≥ 80 years of age (3.5-fold increase in AF risk for every standard deviation reduction in testosterone levels).
Round 2
Reviewer 1 Report
All concerns have been satisfactorily addressed.